# A New Signature That Predicts Progression-Free Survival of Clear Cell Renal Cell Carcinoma with Anti-PD-1 Therapy

**DOI:** 10.3390/ijms24065332

**Published:** 2023-03-10

**Authors:** Jingwei Lin, Yingxin Cai, Yuxiang Ma, Jinyou Pan, Zuomin Wang, Jianpeng Zhang, Yangzhou Liu, Zhigang Zhao

**Affiliations:** Department of Urology & Andrology, Minimally Invasive Surgery Center, Guangdong Provincial Key Laboratory of Urology, The First Affiliated Hospital of Guangzhou Medical University, Guangzhou 510230, China

**Keywords:** clear cell renal carcinoma (ccRCC), immune checkpoint inhibitor (ICI), single-cell RNA-seq, molecular subtype, prognostic model

## Abstract

Immunotherapy has greatly improved the survival time and quality of life of patients with renal cell carcinoma, but the benefits are limited to a small portion of patients. There are too few new biomarkers that can be used to identify molecular subtypes of renal clear cell carcinoma and predict survival time with anti-PD-1 treatment. Single-cell RNA data of clear cell renal cell carcinoma (ccRCC) treated with anti-PD-1 were obtained from public databases, then 27,707 high-quality CD4 + T and CD8 + T cells were obtained for subsequent analysis. Firstly, genes set variation analysis and CellChat algorithm were used to explore potential molecular pathway differences and intercellular communication between the responder and non-responder groups. Additionally, differentially expressed genes (DEGs) between the responder and non-responder groups were obtained using the “edgeR” package, and ccRCC samples from TCGA-KIRC (*n* = 533) and ICGA-KIRC (*n* = 91) were analyzed by the unsupervised clustering algorithm to recognize molecular subtypes with different immune characteristics. Finally, using univariate Cox analysis, least absolute shrinkage and selection operator (Lasso) regression, and multivariate Cox regression, the prognosis model of immunotherapy was established and verified to predict the progression-free survival of ccRCC patients treated with anti-PD-1. At the single cell level, there are different signal pathways and cell communication between the immunotherapy responder and non-responder groups. In addition, our research also confirms that the expression level of PDCD1/PD-1 is not an effective marker for predicting the response to immune checkpoint inhibitors (ICIs). The new prognostic immune signature (PIS) enabled the classification of ccRCC patients with anti-PD-1 therapy into high- and low-risk groups, and the progression-free survival times (PFS) and immunotherapy responses were significantly different between these two groups. In the training group, the area under the ROC curve (AUC) for predicting 1-, 2- and 3-year progression-free survival was 0.940 (95% CI: 0.894–0.985), 0.981 (95% CI: 0.960–1.000), and 0.969 (95% CI: 0.937–1.000), respectively. Validation sets confirm the robustness of the signature. This study revealed the heterogeneity between the anti-PD-1 responder and non-responder groups from different angles and established a robust PIS to predict the progression-free survival of ccRCC patients receiving immune checkpoint inhibitors.

## 1. Introduction

As one of the world’s the most common cancers of the urinary system, it is reported that the number of new cases of renal cancer in the United States in 2022 is estimated to be about 79,000 and the number of deaths is about 13,920 [1]. The main pathological types of renal cell carcinoma are clear cell renal cell carcinoma (ccRCC), papillary renal cell carcinoma, chromophobe cell renal cell carcinoma, and so on. According to the tumor classification of the World Health Organization (WHO) [2] in 2022, ccRCC is the most common pathological type, accounting for approximately 80% of all pathological types [3]. Similar to hepatocellular carcinoma, ccRCC is not sensitive to radiotherapy and chemotherapy. The main treatment methods are surgery-based comprehensive treatment [4,5] including cytokines, molecularly targeted drugs, and ICIs which are aimed at the tumor microenvironment rather than the tumor cells themselves [6]. Immune checkpoint inhibitors (ICI) include anti-PD-1/PD-L1 antibodies (nivolumab, pembrolizumab, atezolizumab, avelumab, etc.) and *CTLA-4* inhibitors (ipilimumab), as well as other drugs [7]. With the emergence of ICIs, the combination of tyrosine kinase inhibitors (TKI) and ICI has become the first-line drug regimen for advanced renal cell carcinoma, which prolongs a patient’s survival and enhances their quality of life.

For a long time, the most commonly used prognostic risk models of renal cell carcinoma included the Memorial Sloan Kettering Cancer Center (MSKCC) risk score [8] and the International Metastatic RCC Database Consortium (IMDC) classification [9,10]. However, on the one hand, these models were developed before the advent of the immunotherapy era, and the accuracy of these models may be affected as ICIs become more and more widely used [11]. On the other hand, patients are stratified mainly by clinicopathological features, which may ignore their molecular features, resulting in uneven response to ICIs. The emergence of ICIs has changed the conventional treatment methods, nevertheless, the beneficiaries of ICIs treatment are still limited [12,13]. Hence, it calls for an imminent need to screen novel biomarkers to identify the most suitable population for ICIs. Nowadays, the predictive markers of immunotherapy mainly focus on PD-L1, tumor mutation load (TMB) [14], microsatellite high instability (MSI-H), DNA mismatch repair deficiency (dMMR) [15], plasma circulating tumor DNA (ctDNA) [16], and so on. Some researchers have also developed the tumor immune dysfunction and exclusion (TIDE) score [17,18], which is a method for predicting ICI treatment response by using a gene expression profile, which is superior to some biomarkers (such as TMB and PD-L1 level). However, there are few biomarkers for PFS in ccRCC treated with immunotherapy. To promote the individualization of ICIs, great importance should be placed on developing novel biomarkers to predict the prognosis and clinical response of ICI treatment.

In our study, by combining single-cell RNA-seq with bulk RNA-seq, the prognostic model was constructed and validated in its ability to predict progression-free survival in ccRCC patients receiving ICI treatment. First of all, seven different types of T cells were identified in the single-cell RNA data of ccRCC treated with anti-PD-1. Then, we characterized the difference in immune characteristics between the response group and non-response group. Specifically, there were stark differences in cell ratio, cell interaction, gene set enrichment, and variation analysis between the two groups. Afterward, differentially expressed genes between responder and non-responder groups were extracted for unsupervised clustering of The Cancer Genome Atlas-Kidney Renal Clear Cell Carcinoma (TCGA-KIRC) (*n* = 533), and two molecular subtypes harboring distinct immune signatures were identified with distinct immune cell infiltration levels, immune checkpoint expression levels, and immunotherapy responsiveness. This was validated in the International Cancer Genome Consortium-Kidney Renal Clear Cell Carcinoma (ICGC-KIRC) (*n* = 91) dataset. Based on these findings, we constructed a robust immunotherapy prognostic signature for ccRCC treated with ICI, with the AUC as high as 0.981 (95% CI: 0.960−1.000) in the training set, the AUC can also reach 0.865 (95% CI: 0.754−0.977) in the validation set. Intriguingly, the signature also successfully distinguishes the responsiveness of ICI treatment. In general, the above results shed some light on the immune characteristics of the ICI treatment response group and non-response group and provide a clinical prognostic model of immunotherapy.

## 2. Results

### 2.1. Cell Subtypes Were Determined by Single-Cell Analysis

Nephrectomy samples from two patients treated with anti-PD-1 antibodies were used to obtain scRNA-seq data, of which one was responsive to nivolumab, and one was nonresponsive. After initial quality control assessment, 27,707 high-quality CD4 + and CD8 + T cells isolated from two distinguished patients were screened and illustrated for further analysis. Principal component analysis (PCA) was used for preliminary dimension reduction of scRNA-seq data. We subsequently applied the UMAP and t-SNE algorithms on the top 30 principal components to visualize the high dimensional scRNA-seq data and 14 clusters were obtained. Previous canonical cell markers [19,20,21] and CellMarker 2.0 [22] helped us to identify seven T cell subsets (Figure 1A,E). Namely, exhausted CD8 + T cells, cytotoxic CD8 + T cells, resident memory CD4 + T cells, cluster1-resident memory CD8 + T cells, cluster2-resident memory CD8 + T cells, CD4 + CD25 + FOXP3 + regulatory T cells, and naive CD4 + T cells. Abbreviated as EX CD8 + T, Cytotoxic CD8 + T, RM CD4 + T, C1−RM CD8 + T, C2−RM CD8 + T, Treg CD4 + T, and Naive CD4 + T, respectively. Figure 1B,D illustrates the cell composition between 2 samples and a heatmap of the top five marker genes expression in 14 clusters, the proportion of Treg CD4 + T in the non-responsive group was significantly higher than that in the responsive group. It can be seen from Figure 1E that, unlike C1-RM CD8 + T cells, C2-RM CD8 + T cells overexpressed exhausted T cell markers (*PDCD1*, *TIM3/HAVCR2*, *TOX2*, *LAG3*, *CD200R1*), however, the expression level was lower than that of EX CD8 + T cells. Likewise, immune checkpoint enrichment analysis showed that EX CD8 + T had the highest score, followed by C2-RM CD8 + T, and C1-RM CD8 + T had the lowest score (Figure 1C), The checkpoint genes used for enrichment analysis are shown in Appendix A. According to the absolute value of log2FC ≥ 0.5 and adjusted *p* < 0.05, 432 differentially expressed genes were identified from the responsive and non-responsive groups, including 281 up-regulated genes and 151 down-regulated genes (Appendix A).

### 2.2. The Functional Pathway and Cell Interactions of Classified T Cell Clusters

The main markers of RM T cells include *CD103/ITGAE* and *CD69* (Figure 1E). Similar to exhausted T cells, studies have also revealed that some RM T cells express high levels of immune checkpoint markers (such as *PD-1/PDCD1, CTLA-4*, *TIM-3/HAVCR2, LAG-3*), but the difference is that those RM T cells will restore strong immune function when re-exposed to the appropriate antigens [23,24,25]. The violin diagrams (Figure 2A) illustrate the distribution of immune checkpoint genes in different T cell subsets, indicating that *PD-1/PDCD1*, *TIM-3/HAVCR2*, *LAG-3*, and *CTLA-4* are highly expressed in EX CD8 + T and C2-RM CD8 + T, which suggests that these two types of cells may be the main target of immune checkpoint inhibitors. The GSVA algorithm provides a great method for us to elucidate the underlying biological characteristics of specific single-cell subsets in responsive and non-responsive groups. We focused on exhausted T cells and C2-RM T cells with rich immune checkpoint gene expression in the responsive and non-responsive groups. As depicted in Figure 2B, in the exhausted T cell subsets, the Notch signaling pathway, mTOR signaling pathway, and oxidative phosphorylation were predominantly enriched in the responsive group, whereas the PPAR signaling pathway, P53 signaling pathway, and MAPK signaling pathway were enriched in the non-responsive group. When it comes to that C2-RM T subsets, KRAS signaling, IL2_STAT5 signaling, P53 pathway, MTORC1 signaling, TGF-β signaling [26], and MYC targets v2 pathway were concentrated in the non-responsive group, however, the Notch signaling pathway and oxidative phosphorylation were mainly enriched in the responsive group (Figure 2C).

The CellchatDB algorithm is also used to reveal the connections and differences in intercellular communication between the responsive group and the non-responsive group, the bar chart Figure 2D displays the relative expression intensity of specific signaling pathways in the response group and non-response group. In addition, we also characterize the heatmap of communication among cell subsets of the two groups. Intriguingly, contrary to the response group, more and stronger cellular connections were observed between Treg CD4 + T cells and other cell subsets (Figure 2E). A plausible explanation would seem to be that Treg CD4 + T cells may play a negative role in regulating immune response by secreting inhibitory cytokines in various immune cell subsets, which may lead to the immune escape of tumor cells [26,27].

### 2.3. The Identification of Two Molecular Clusters

DEGs identified between the responsive group and the non-responsive group were selected as candidate markers for the unsupervised clustering of TCGA-KIRC (*n* = 533), and finally, two molecular subsets were identified (Figure 3A), K-M survival analysis showed that cluster 1 was significantly better than cluster 2 (*p* = 0.022) (Figure 3B). Then, we analyzed the clinicopathological features of C1 and C2 including T, N, M, stage, and gender, and concluded that there were significant differences in clinicopathological features except for N stage (*p* = 0.086) between C1 and C2. Specifically, the T, M, and stage of C2 are higher than those of C1, which is also consistent with survival analysis, and it is worth noting that the proportion of males in C2 is also higher than that in C1 (Figure 3C).

### 2.4. Distinct Immune Characteristics between Two Clusters

There are significant differences in clinicopathological features between C1 and C2. More importantly, we discussed the immune characteristics between C1 and C2 from the aspects of immune cell infiltration abundance, immune checkpoint expression, TIDE, and dysfunction score, and found a conspicuous distinction in immunotherapy responsiveness between C1 and C2. First of all, the CIBERSORT algorithm was adopted to estimate the infiltration abundance of 22 immune cells in C1 and C2. Afterward, we used the Wilcoxon rank-sum test to explore whether there was a difference in the expression of immune cells between the two groups. The results demonstrated that the infiltration levels of many kinds of immune cells were notably different between C1 and C2 (Figure 4A). Particularly, great importance was attached to the infiltration levels of CD8 T cells and T cells regulatory (Tregs), which were significantly higher in C2 than those in C1. High expression of regulatory T cells in C2 may indicate poor immunotherapy benefits [28]. Combined with the bar chart showing immune checkpoint expression (Figure 4B), where most canonical markers of exhausted CD8 + T including *TIM3/HAVCR2*, *TOX2*, *LAG3*, and *CD200R1* are highly expressed in C2, we could conclude that the abundance of exhausted CD8 T cells infiltration in C2 may be higher than that in C1, providing a new direction for immunotherapy. Furthermore, the TIDE scores and dysfunction scores were calculated to evaluate the responsiveness of immunotherapy between the two groups (Figure 4C). The score of C1 is lower than that of C2, implying that C1 is more likely to benefit from immunotherapy. Similarly, the unsupervised clustering algorithm is performed again in the ICGC-KIRC cohort (*n* = 91) with the same candidate genes and two clusters were also obtained (Figure 4D), whose TIDE scores and dysfunction scores were also obviously distinct as shown in Figure 4E.

### 2.5. Relationship among PDCD1 Expression, Anti-PD-1 Response, and Survival

In theory, the higher the expression of PD-L1, the higher the effectiveness rate of immunotherapy should be, yet this was not the case. We analyzed the relationship between *PD-1/PDCD1* expression and immunotherapy response as well as survival in ccRCC patients receiving anti-PD-1 therapy in the CheckMate cohort derived from Braun et al. In our study, patients were divided into a high *PD-1/PDCD1* expression group and a low *PD-1/PDCD1* expression group according to the median expression level. Figure 5A,B showed that there was no significant difference in overall survival and progression-free survival between the two groups (*p* = 0.12), and there was no significant difference of the anti-PD-1 treatment response between the two groups (*p* = 0.726). Similar results were also found in the Ascierto et al. cohort and the Shiuan et al. cohort with *p* values of 1.000 and 0.576, respectively (Figure 5C). Our results also indicate that overall survival (Figure 5D) and progression-free survival (Figure 5E) were better in an anti-PD-1-responsive group than in the non-responsive group with a *p* value less than 0.001. In conclusion, the expression of PD1/PDCD1 alone cannot accurately predict the response and prognosis of anti-PD-1 therapy, and it is extremely necessary to predict the survival and responsiveness of anti-PD-1 therapy in an accurate and timely manner.

### 2.6. PIS Predicts Survival and Response to Anti-PD-1 Therapy Precisely

The CheckMate cohort (*n* = 172) with complete and clear clinicopathological information was randomly assigned to the training (*n* = 121) and validation cohorts (*n* = 51) at a 7:3 ratio. To begin, 1666 candidate genes were screened by a univariate Cox regression algorithm, to select genes significantly related to PFS (*p* < 0.05). According to the selection criteria, 35 PFS-associated genes with *p* < 0.05 were screened out for the LASSO Cox regression algorithm to ensure the robustness of the prognostic model (Figure 6A). Afterward, the lambda.min was determined as the optimal lambda value by tenfold cross-validations, and the above 27 prognostic genes with non-zero coefficients were all enrolled (Figure 6B). Ultimately, multivariate analysis and a stepwise algorithm were used to ensure that the Akaike information criterion (AIC) is the minimum, thus a prognostic immune signature (PIS) consisting of 18 genes was constructed to predict the PFS of anti-PD-1 therapy in the treatment ccRCC.

Based on the coefficients, the signature was confirmed. Figure 7A was made to display the details of 18 genes used to construct PIS, including risk coefficient, *p* value obtained by multivariate regression analysis, and hazard ratio. According to the median cutoff value of the PISs, patients were separated into high-risk and low-risk groups. Compared with the low-risk group, Figure 6C illustrates that the low-risk group with lower PISs had better PFS (*p* = 3.998 × 10^−13^). The marked difference in survival between the high-risk group and low-risk group was also observed in the validation set (Figure 6D). We further explored the difference between the high-risk group and the low-risk group in response to anti-PD-1. Intriguingly, consistent with the previous analysis, the low-risk group had higher response rates in both the training set (Figure 6E) and the verification set (Figure 6F), with *p* values of 0.002 and 0.008, respectively.

Then, the potential accuracy and robustness of the model were further evaluated via the “timeROC” package in the training cohort, with 1-, 2- and 3-year AUCs of 0.940 (95% CI: 0.894−0.985), 0.981 (95% CI: 0.960−1.000), and 0.969 (95% CI: 0.937−1.000), respectively (Figure 7B). The AUCs of 1-, 2- and 3-years were 0.816 (95% CI: 0.681−0.950), 0.865 (95% CI: 0.754−0.977), and 0.827 (95% CI: 0.699−0.955) in the validation set, respectively, (Figure 7C), also demonstrating exceptional predictive potential, especially for the PFS within 3 years.

## 3. Discussion

The medical treatment of clear cell renal cell carcinoma has experienced the cytokine era, targeted therapy era, and immune checkpoint inhibitor era. The advent of ICI has changed many traditional cancer treatment methods and greatly enriched the treatment options for advanced renal cell carcinoma [6]. Yet the beneficiaries of ICI treatment are still limited due to the complexity of the tumor microenvironment and the fact that the rate of side effects caused by ICI therapy is also high [32,33]. With the advent of the era of precision medicine, it may not be feasible to rely on traditional stratification to determine who is suitable for immunotherapy [8,9,10]. It is necessary to develop new predictive biomarkers to screen out the most likely beneficiaries. By integrating single-cell data and bulk-RNA sequencing data, this study aims to construct a prognostic immune signature (PIS), which can accurately predict the progression-free survival (PFS) of patients with ccRCC with anti-PD-1 therapy. Combined with the ROC curve, PIS can serve as a reliable signature to predict the prognosis and responsiveness of patients receiving immunotherapy. Meanwhile, this study also depicts the immune characteristics of the responder and non-responder from the single-cell level.

In this study, we re-recognized and clustered CD4 + T and CD8 + T cells from the responder and non-responder. According to the canonical markers *CD103/ITGAE* and CD69 of tissue-resident memory (TRM) cells [34], we identified two different states of TRM CD8 + T cells called C1-RM CD8 + T cells and C2-RM CD8 + T cells, respectively. *ITGAE* + TRM play an important role in immune surveillance and the restriction of tumor growth of solid tumors [35]. Studies have shown that the existence of TRM cells in tumors is related to the improvement of the survival time of hepatocellular carcinoma [36], breast cancer [37], ovarian cancer [38], cervical cancer [39], and bladder cancer [40]. In addition, published articles have shown that TRM in tumors can also predict immunotherapy responses due to the expression of immune checkpoint genes [34]. Intriguingly, consistent with previous studies, we found that a C2-RM CD8 + T cell population enriches immune checkpoint genes PDCD1, TIM3/HAVCR2, TOX2, LAG3, and CD200R1, which are similar to exhausted T (Tex) cells and may be targeted for immunotherapy (Figure 1C,E). Strikingly, almost no immune checkpoint expression was found in the C1-RM CD8 + T cell population, suggesting that not all RM cells respond to immunotherapy.

In addition, GSVA results in single-cell sequencing suggested that KRAS signaling pathways and MAPK signaling pathways were enriched in non-respondents (Figure 2B,C). Basic studies have revealed that the KRAS signal can promote immune escape by regulating the stability of the PD-L1 mRNA 3UTR region and up-regulating PD-L1 expression [41]. In addition, as a KRAS signal inhibitor, AMG-510 was shown to promote anti-tumor immunity by inhibiting the PD-L1 signal [42]. The above results indicate that KRAS signaling is closely related to tumor immune escape, and KRAS signal enrichment may indicate poor immunotherapy benefits (Figure 2B). Similarly, tumor cells can also acquire immune tolerance through MAPK, WNT, CDK4/6, and PTEN signaling pathways. For instance, the MAPK signaling pathway inhibits the activity of immune cells in the tumor microenvironment by up-regulating the release of IL-6/IL-10 and other cytokines. In addition, a BRAF-V600E mutation in the MAPK pathway can lead to the increase of drug resistance in tumor cells [43,44], which is consistent with our results, that is, that the MAPK signaling pathway is enriched in the non-responder (Figure 2C).

PD-1/PD-L1 is one of the earliest markers used to predict the therapeutic response to ICI. It is a useful but imperfect biomarker for predicting anti-PD-1/PD-L1 antibodies. One of the reasons is that its clinical efficacy varies greatly according to different cancer types and treatment schemes [45]. Recently, researchers analyzed the efficacy of ICIs in patients with gastric cancer with low PD-L1 expression. The results showed that the treatment scheme of immunotherapy combined with chemotherapy did not bring significant benefits to patients’ overall survival (OS) and progression-free survival (PFS) compared with the chemotherapy alone [46]. On the contrary, another study showed that in the first-line treatment of advanced esophageal squamous cell carcinoma, the curative effect of anti-PD-1 antibody combined with chemotherapy in people with low PD-L1 expression was still significantly better than that of chemotherapy alone [47], which added strong evidence in favor of the application of combined therapy in patients with low PD-L1 expression, and also demonstrated that the relationship between PD-1/PD-L1 expression level and the anti-PD-1 curative effect was not consistent. In the phase III clinical trial investigating the treatment of advanced metastatic renal cell carcinoma patients with PD-1 inhibitor nivolumab, researchers used the PD-L1 protein expression of tumor cells as a predictive molecular marker, which showed that patients can still benefit from immunotherapy when the expression level of PD-L1 is lower [48]. In our study, we initially explored the relationship between PDCD1 expression and ICI response. Unfortunately, no significant association was found in three independent anti-PD-1 immunotherapy cohorts (Figure 5C), and there was no remarkable relationship between PDCD1 expression and the overall survival or progression-free survival of ccRCC patients treated with ICI (Figure 5A). TMB was first considered as a potential biomarker for ICI treatment of melanoma [49] and then gradually extended to other types of cancers including non-small cell lung cancer [50], and head and neck squamous cell carcinoma [51]. Although TMB is easy to quantify, its predictive clinical value should not be exaggerated. Mary et al.’s research results suggest that TMB is a reasonable predictor of immunotherapy for non-small cell lung cancer, but it does not support TMB as a predictive biomarker for immunotherapy for RCC alone. The relationship between PD-L1 expression and TMB is intriguing. Previous studies have elucidated that PD-L1 expression has nothing to do with TMB in advanced lung cancer patients treated with anti-PD-1 [52,53]. There is no significant correlation between PD-L1 expression and TMB in most cancer subtypes [54], which also indicates that if PD-L1 expression is combined with TMB in the clinic, a stronger prediction of the effectiveness of ICI treatment could be obtained.

The PIS signature consisting of 18 genes can accurately predict PFS in ccRCC cases with anti-PD-1 therapy and exhibit significant survival differences between the low-risk group and high-risk group. The AUC values of 1-year, 2-year, and 3-year PFS were 0.940 (95% CI: 0.894–0.985), 0.981 (95% CI: 0.960–1.000), and 0.969 (95% CI: 0.937–1.000), respectively. Although numerous researchers have developed many biomarkers to predict the survival time of patients with ccRCC, they have not been able to predict the survival time of patients receiving immunotherapy because the patients included did not receive anti-PD-1 treatment [55,56,57]. There are also quite a few biomarkers that can predict the responsiveness of ccRCC to anti-PD-1, for example, a novel signature composed of 47 genes can predict the response to anti-PD-1 therapy, and the AUC value is as high as 0.93 [58]. Long et al. developed a mutation-based gene set to predict immunotherapy results, which provides a good direction for accurate immunotherapy for ccRCC [59]. One thing we need to call attention to is the fact that, although immunotherapy patients may achieve a complete remission or partial remission, residual cancer cells may still lead to disease progression or even death, so progression survival time may be a more objective indicator of immunotherapy benefits. Thus, the PIS was developed to predict progression-free survival with anti-PD-1 therapy. Of course, our research also has some shortcomings. First, we merely built and validated the model with a limited sample size. More samples are still needed to support it, although the AUC value is above 0.9. Secondly, it is still necessary to clarify the molecular mechanism of the way that the 18 genes constituting PIS affect the prognosis of immunotherapy patients in vivo and in vitro functional experiments.

## 4. Materials and Methods

### 4.1. Data Collection and Processing

We collected and processed 6 independent public datasets in this study, consisting of single-cell RNA-seq cohorts [60], bulk RNA-seq cohorts, and clinical cohorts treated with anti-PD-1 antibodies [29,30,31]. The scRNA-seq data were obtained from two patients who underwent cytoreductive nephrectomy after anti-PD-1 treatment (one was responsive to nivolumab, and the other was nonresponsive). The original single cell sequencing data included 31,184 T cells, consisting of 11,193 T cells in the non-responder and 19,991 T cells in the responder. We downloaded and re-analyzed the scRNA-seq data from the University College London (UCL) website (accessed on 13 October 2022: https://rdr.ucl.ac.uk/articles/dataset/Multi-region_scRNA_and_scTCR_data_on_ADAPTeR_patient_cohort/16573640/1). The harmony algorithm was used to remove batch effects between samples. Clinicopathological information for all patients is detailed in the original literature [60]. The TCGA-KIRC cohort (*n* = 533), ICGC-KIRC cohort (*n* = 91) with bulk RNA-seq data and corresponding clinical information were retrieved from the cBioportal database (accessed on 21 October 2022: https://www.cbioportal.org/) and ICGC (accessed on 21 October 2022: https://dcc.icgc.org/search), respectively. The clinical treatment cohort obtained from Braun et al., namely the CheckMate cohort, contains 39 responders, 133 non-responders, and 9 not evaluated (NE) patients. It is worth noting that the CheckMate cohort derives from 3 prospective randomized clinical trials (CheckMate 009: NCT01358721, CheckMate 010: NCT01354431, and CheckMate 025: NCT01668784) of anti-PD-1 antibodies in the treatment of advanced ccRCC. In detail, the CheckMate cohort was an integrated cohort of the above three randomized clinical trial cohorts selected against certain criteria, leaving 181 nivolumab-treated samples after the removal of the everolimus-treated samples. Relevant information has been described in detail in the original research [48,61,62]. Additionally, two other independent datasets of anti-PD-1 treatment were also downloaded to explore the relationship between PDCD-1 expression and response to anti-PD-1 antibodies; the Ascierto et al. immunotherapy cohort (*n* = 11) (7 non-responders, 4 responders), which was available from Gene Expression Omnibus (GEO) database (accessed on 25 October 2022: https://www.ncbi.nlm.nih.gov/geo/) (accession no.GSE67501), and the Shiuan et al. cohort (*n* = 15) (7 non-responders, 8 responders), which was available on the supplementary table of the corresponding article [31]. Genes mapped to multiple probes were calculated by their average values. The workflow of the study is presented in Figure 8.

### 4.2. Single-Cell RNA-Seq Analysis

The Seurat package (v 4.2.1) was utilized to generate the object for further analysis. Firstly, we performed standard data preprocessing, in which we calculated the percentage of the gene numbers, cell counts, and mitochondria sequencing count. Genes with less than 3 cells detected and disregarded cells with less than 50 detected gene numbers were excluded. Then, cells with fewer than 500 or more than 4000 detected genes and those with a high mitochondrial content (>10%) were filtered out. Ultimately, 27,707 cells were retained for downstream analysis. To normalize the library size effect in each cell, we scaled UMI counts using scale.factor  =  10,000. The top 30 principal components were kept for t-distributed stochastic neighbor embedding (t-SNE) and Uniform Manifold Approximation and Projection for Dimension Reduction (UMAP) visualization and clustering. We performed cell clustering using the “FindClusters” function (resolution  =  0.6) implemented in the Seurat R package. Afterward, seven types of CD4 + and CD8 + T cells were identified according to previous canonical markers. Moreover, we utilized the “FindMarkers” function to calculate differentially expressed genes between the responder and non-responder cells with the filter value of absolute log2 fold change (logFC) ≥ 0.5 and adjusted *p*-value < 0.05 [63,64,65,66].

Meanwhile, the Seurat package and AddModuleScore approach were used to accomplish the enrichment analysis of the immune checkpoint gene set derived from the original study. In addition, based on 50 hallmark and 186 KEGG gene sets, Gene Set Variation Analysis (GSVA) was performed to explore differences in functional pathways between specific T cells with the GSVA package (v1.46.0) of R software with default parameters [67]. The hallmark and KEGG gene sets were downloaded from the Molecular Signatures Database (MSigDB) (accessed on 1 November 2022: https://www.gsea-msigdb.org/gsea/msigdb/index.jsp), a joint project between UC San Diego and the Broad Institute [68,69]. Importantly, after passing quality control, the CellChat (v1.6.0) package was used to explore the cellular communication differences between the responder and non-responder cells [70].

### 4.3. Subgroup Recognition Based on Consistent Clustering

An unsupervised clustering algorithm was performed using the ConsensusClusterPlus package (v1.62.0) to identify subtypes with different biological characteristics in the TCGA-KIRC cohort (*n* = 533) [71]. Candidate genes were 432 differential genes between the responder and non-responder clusters. The parameters of consensus clustering were set to the following values under the algorithm: maxK = 5, reps = 50, pItem = 0.8, pFeature = 1. Empirical cumulative distribution function (CDF) plots displayed consensus distributions for each k. The delta area score (*y*-axis) indicated the relative increase in cluster stability. The above algorithm was also utilized for unsupervised clustering in the ICGC-KIRC cohort (*n* = 91). Combining the cumulative distribution curve, delta area curve, and clustering consistency histogram, the optimal number is judged.

### 4.4. Immune Features Analysis between Two Subgroups

To explore the potential biological differences between the two subgroups, we first analyzed the clinicopathological features of the two subgroups, including gender, T, N, M, stage, and survival time. Most importantly, we tried to explore the differences in immune characteristics between the two groups. Firstly, the proportions of the 22 immune cells between the two subgroups were estimated via the CIBERSORT algorithm [72]. Only cases with a CIBERSORT output of *p*  <  0.05 were considered to be eligible for subsequent analysis and visualization. Furthermore, to confirm the difference in immune characteristics between the two clusters, 39 immune checkpoint genes (ICG) from previous studies were enrolled to analyze the difference in ICG expression between the two clusters. The Wilcoxon rank-sum test was used to compare the differences in immune cell infiltration and ICG expression between the two subgroups. Additionally, the TIDE score and T-cell dysfunction score calculated online (http://tide.dfci.atherard.edu/) were used to evaluate the responsiveness of immunotherapy between the two groups [18,73]. The lower the score, the more likely the group is to benefit from immunotherapy. The TIDE and dysfunction scores were performed again in the ICGC-KIRC cohort (*n* = 91).

### 4.5. Construction and Validation of the Prognostic Immune Signature

By comparing cluster 1 with cluster 2, 1666 GEGs were identified (absolute logFC ≥ 1.0 and *p*-value < 0.05). A total of 172 samples originate from independent randomized clinical trial cohorts (CheckMate 009, CheckMate 010, and CheckMate 025) with complete gene expression and exact clinical information which was defined as the CheckMate cohort, which is used to construct and verify the prognosis model of immunotherapy. To begin with, the CheckMate cohort (*n* = 172) was randomly assigned to the training group (*n* = 121) and validation cohort (*n* = 51) at a 7:3 ratio with the createDataPartition function. DEGs between the two subgroups were analyzed by univariate Cox to obtain candidate prognostic genes (*p* < 0.05), Afterward, the least absolute shrinkage and selection operator (LASSO) method performed through the “glmnet” (v4.1.4) package was used to minimize overfitting risk [74] and select the optimal gene combination with the lowest Akaike information criteria (AIC) in a stepwise algorithm. The prognostic immune signature (PIS), including 18 markers, was ultimately constructed according to the regression coefficient derived from the multivariate Cox regression model and the optimized genes to precisely forecast survival and response to anti-PD-1 antibodies in ccRCC. The formula is $prognostic\;immune\;signature\;score\;\left(PISs\right)=\sum_{i=1}^{n}\left(\beta_{i}*\mathrm{Exp}i\right)$, where *β_i_* represents the coefficient of the marker, Expi was the candidate marker’s expression level, and *n* denotes the number of the incorporated markers.

Then, based on the median PISs, patients were classified into high- and low-risk groups, and the Kaplan–Meier plot was applied to evaluate differences between the high-risk and low-risk subgroups by the R package “survival” (v3.4.0) [75] in the training cohort and validation cohort, respectively. To evaluate the efficiency and robustness of the PISs, the receiver operating characteristic (ROC) curve performed by the R package “timeROC” (v0.4) was used to predict 1-, 2-, and 3-year survival. In addition, we further identified the distinct responses of high-risk and low-risk groups to anti-PD-1 therapy in the training set and the validation set, respectively.

### 4.6. Statistical Analyses

The statistical analyses and visualization were conducted using the R software (v4.2.2). Differences between variables were analyzed using independent t-tests, chi-square tests, Fisher’s exact test, rank-sum test, or ANOVA tests. All statistical tests were two-sided. *p* < 0.05 was considered statistically significant.

## 5. Conclusions

In the present study, a robust signature composed of 18 genes predicting PFS in ccRCC with anti-PD-1 therapy was developed. Additionally, we also explored the potential biological characteristics of anti-PD-1 therapy responders and non-responders from the perspective of the single-cell data level also revealed the molecular pathway differences between them.

## Figures and Tables

**Figure 1 ijms-24-05332-f001:**
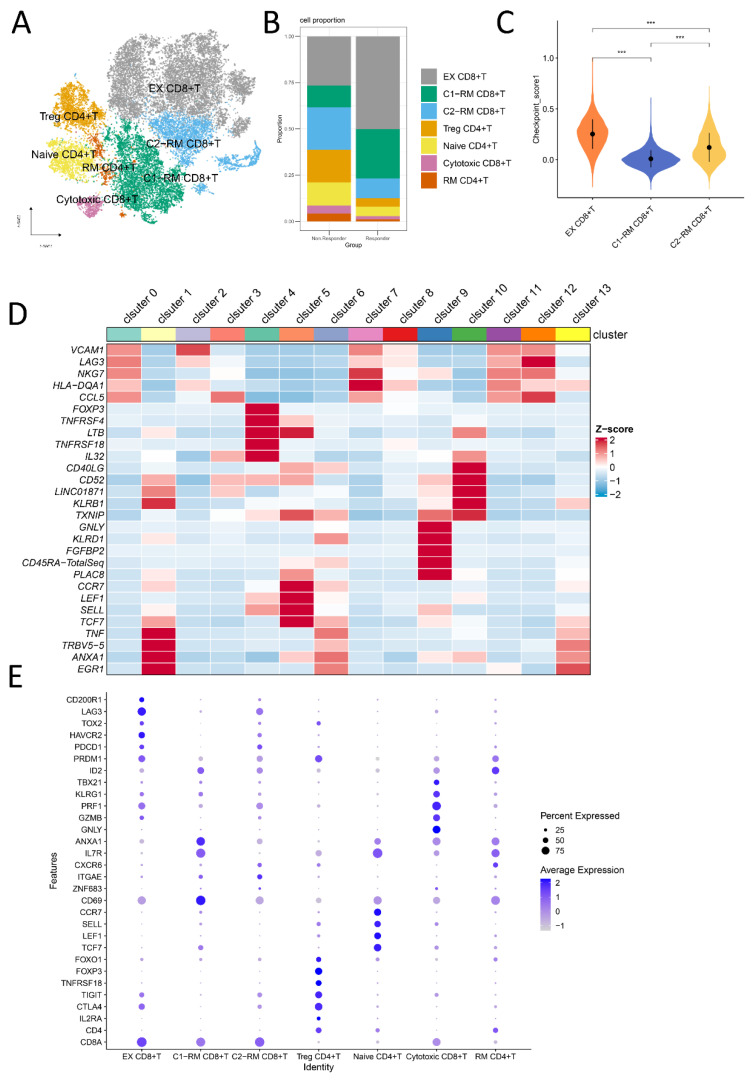
Single-cell RNA-seq profiling of anti-PD-1 responder and non-responder. (**A**): 27,707 high-quality CD4 + T and CD8 + T cells were classified into seven types of cells via the t-SNE dimensionality reduction algorithm based on canonical cell markers. (**B**): The stacking histogram displayed the difference in the distribution of seven types of T cells between the responder and non-responder. (**C**): Enrichment analysis of immune checkpoint genes of C1-RM CD8 + T, C2-RM CD8 + T, and EX CD8 + T clusters, indicating that EX CD8 + T had the highest score, followed by C2-RM CD8 + T, and C1-RM CD8 + T had the lowest score (*** means *p* < 0.001) (**D**): Heatmap depicting expression of top five marker genes among 14 clusters, the color represents the averaged scaled expression value. (**E**): Canonical markers of seven different cell types (exhausted CD8 + T cells, cluster1-resident memory CD8 + T cells, cluster2-resident memory CD8 + T cells, CD4 + CD25 + FOXP3 + regulatory T cells, naive CD4 + T cells, cytotoxic CD8 + T cells, and resident memory CD4 + T cells) are displayed in the bubble diagram, in which dot size and color represent the percentage of marker gene expression and the averaged scaled expression value, respectively.

**Figure 2 ijms-24-05332-f002:**
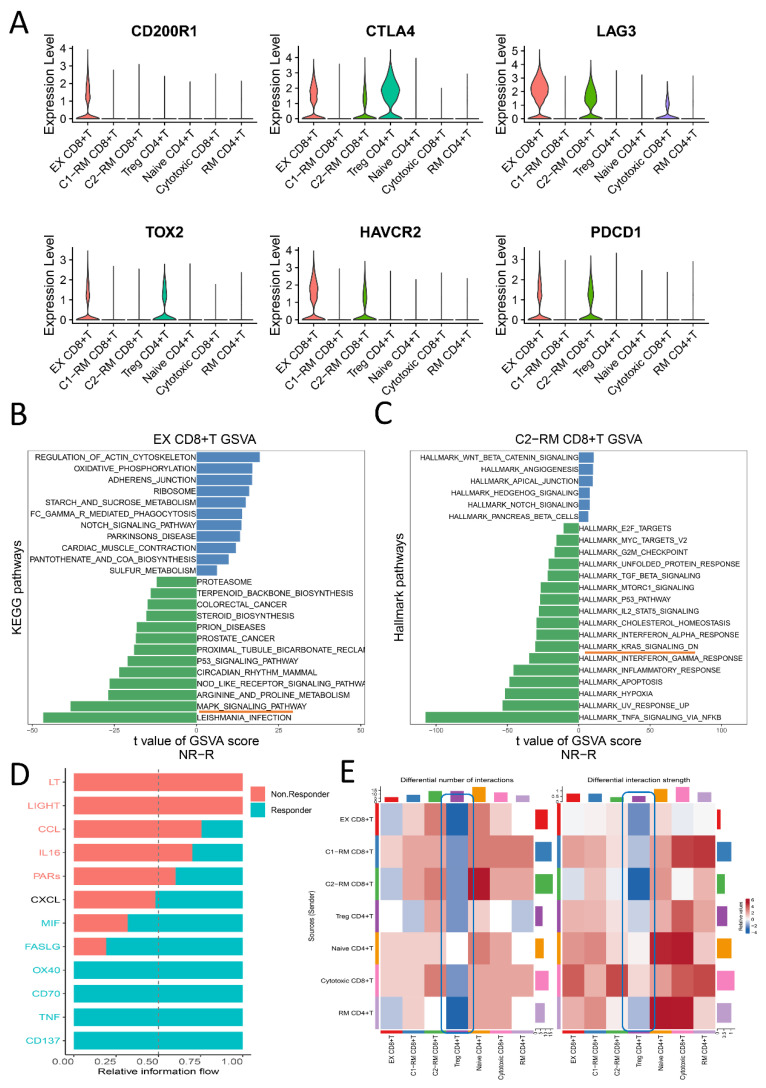
Differences in molecular pathways and cellular communication between the responder and non-responder groups. (**A**): Violin diagram displays the distribution of six immune checkpoint genes (*PDCD1*, *TIM3/HAVCR2*, *TOX2*, *LAG3*, *CD200R1*) in seven types of T cell subsets. Different colors indicate different cell populations. (**B**,**C**): Analysis of gene set variation of EX CD8 + T cells and C1-RM CD8 + T cell subsets in the responder and non-responder groups, the differential KEGG pathways and hallmark pathways between them were sorted according to the *t* value of statistical results. The orange underline indicates that the MAPK signaling pathway was enriched in the non-responsive group, while KRAS signaling was concentrated in the non-responsive group. The blue bar indicates the responder while the green denotes the non-responder. (**D**): Bar graph illustrating the relative information flow of the signaling pathway between the responder and non-responder groups, blue indicates that the corresponding pathway is highly enriched in respondents, and vice versa. (**E**): Heatmap showing the difference in the number and strength of cell interactions. The blue rectangle indicates that, contrary to the response group, more and stronger cellular connections were observed between Treg CD4 + T cells and other cell subsets. The darker the blue color, the greater the strength and number of cell interactions in the non-responsive group.

**Figure 3 ijms-24-05332-f003:**
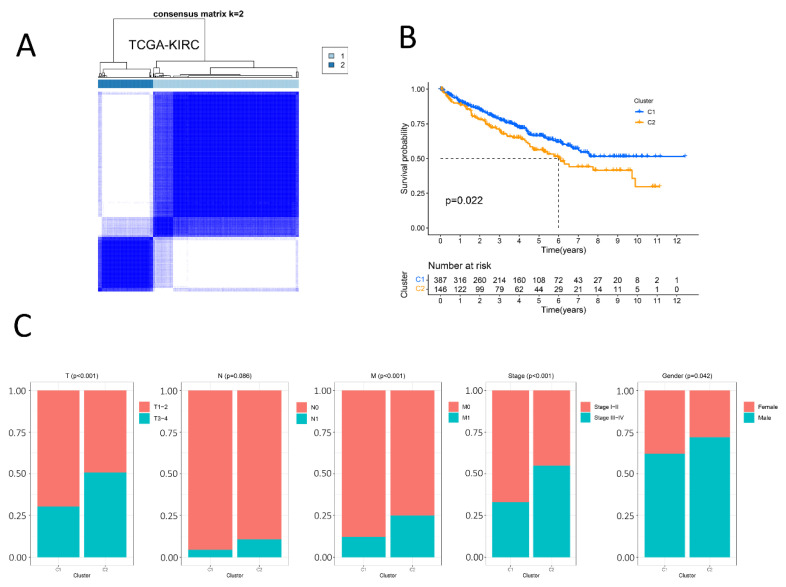
Identification of two heterogeneous molecular clusters with distinct clinical features by consensus clustering analysis. (**A**): Consistent clustering is performed in The Cancer Genome Atlas- Kidney renal clear cell carcinoma (TCGA-KIRC) cohort and two clusters are obtained. Two clearly delimited blue squares represent different molecular subtypes. (**B**): Kaplan–Meier survival plot of the two clusters in the TCGA-KIRC cohort. The blue color indicates cluster 1, and the orange represents cluster 2 (**C**): The proportion of clinicopathologic features (T, N, M, stage, gender) of the two subtypes. There exist significant differences, except for the N stage (*p* = 0.086), between the two clusters. *p* < 0.05 was considered to be statistically significant.

**Figure 4 ijms-24-05332-f004:**
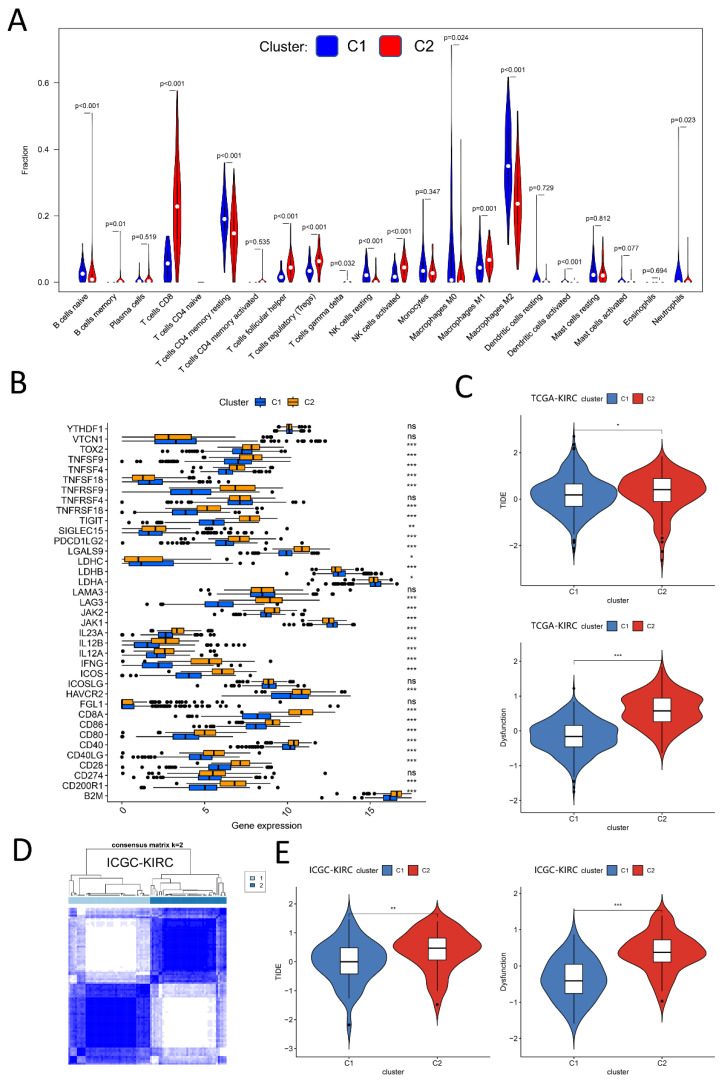
Distinct immune characteristics between two clusters. (**A**): The difference of 22 types of immune cell infiltration between cluster 1 and cluster 2. (**B**): Distribution difference of 39 immune checkpoint genes between two clusters. (**C**): Difference of tumor immune dysfunction and exclusion (TIDE) score and dysfunction score between two clusters in TCGA-KIRC cohort. The lower the scores of TIDE and dysfunction, the more likely patients are to benefit from immunotherapy. (**D**,**E**): Consistent clustering in (ICGC-KIRC) cohort to get two clusters with different TIDE and dysfunction scores. (ns, not significant; * represents *p* < 0.05; ** denotes *p* < 0.01; *** means *p* < 0.001).

**Figure 5 ijms-24-05332-f005:**
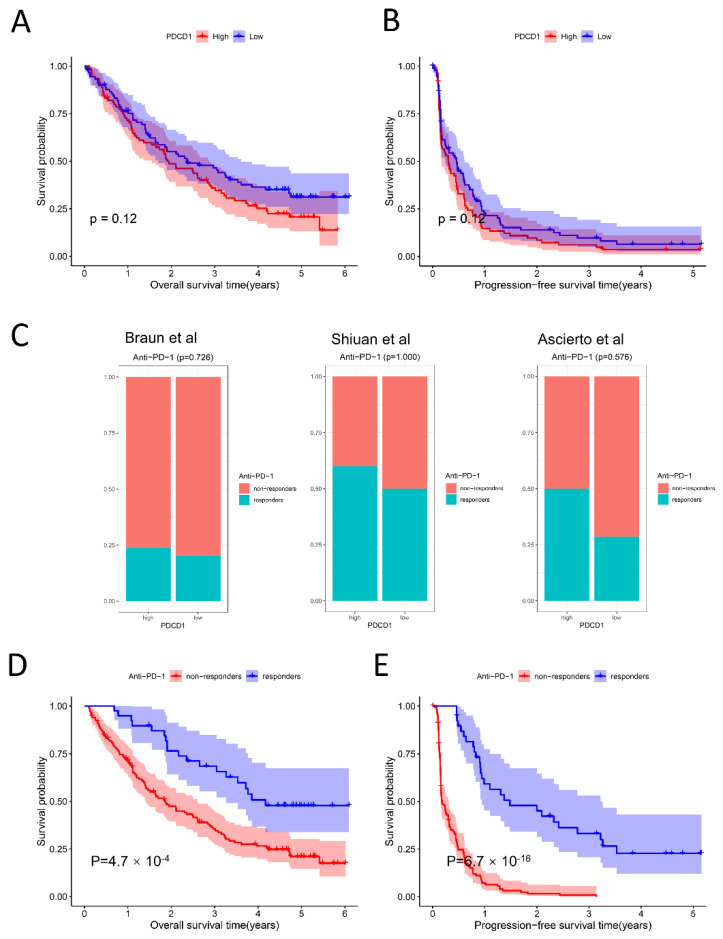
Relationship among *PDCD1/PD-1* expression, anti-PD-1 response, and survival. (**A**,**B**): Correlation between *PDCD1* expression and overall survival (left) as well as progression-free survival (right) based on Kaplan-Meier analysis in CheckMate cohort (*n* = 181) (*p* = 0.12). The patients were dichotomized at the median *PDCD1* expression. (**C**): Correlation between *PDCD1* expression and anti-PD-1 therapeutic responsiveness in three independent immunotherapy cohorts [29,30,31]. (**D**,**E**): Survival analysis of overall survival and progression-free survival between anti-PD-1 responsive and non-responsive groups in the CheckMate cohort (*n* = 172). A total of nine patients who were not evaluated after immunotherapy were excluded from 181 patients. *p* < 0.05 was considered to be statistically significant.

**Figure 6 ijms-24-05332-f006:**
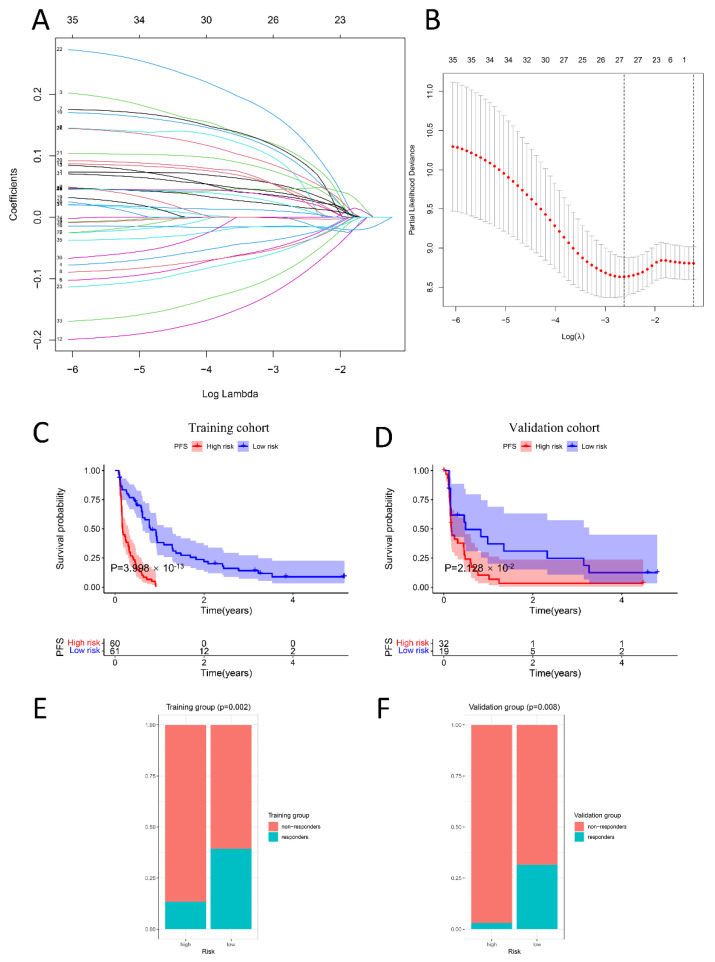
Construction and validation of the prognosis model for PFS in the CheckMate cohort (*n* = 172). (**A**,**B**): A total of 27 genes correlated with PFS were selected via LASSO regression analysis and ten-fold cross-validations for screening of the optimal parameter lambda. (**C**,**D**): The samples were randomly classified into training and validation cohorts at a 7:3 ratio with the createDataPartition algorithm. Kaplan-Meier survival plots suggested that the low-risk group exhibited better PFS than the high-risk group in the training cohort and validation cohort. Blue indicates the survival curve of the low-risk group, and red indicates the survival curve of the high-risk group (**E**,**F**): There existed obvious differences in the proportion of responders and non-responders between the high-risk group and low-risk group both in the training cohort and validation cohort. The proportion of responders in the low-risk group is higher. The ratio of orange to green means the ratio of responders to non-responders.

**Figure 7 ijms-24-05332-f007:**
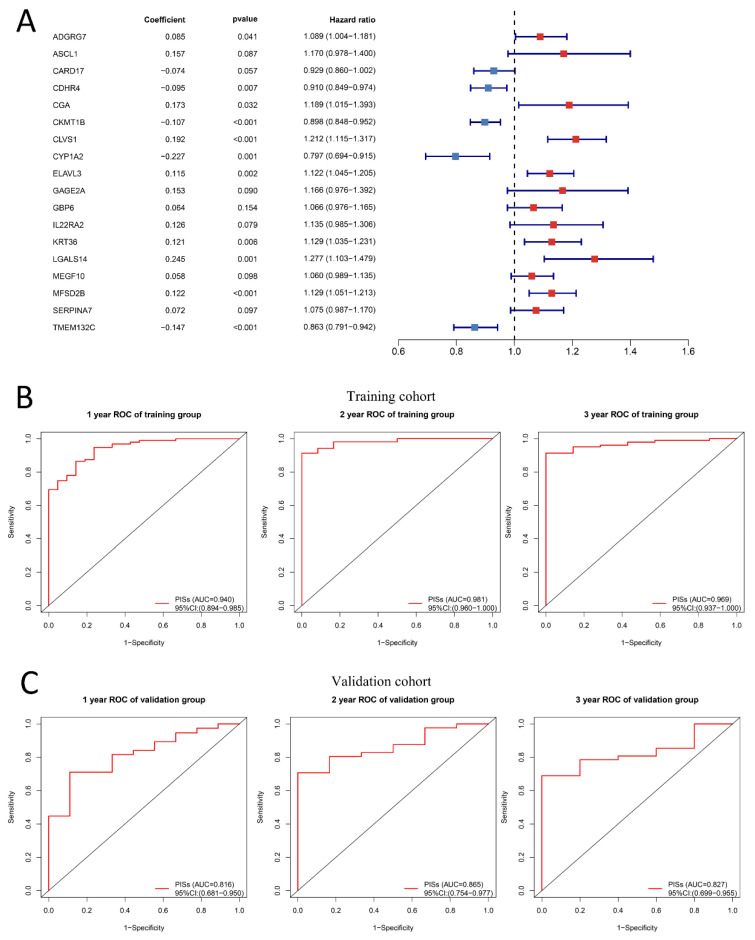
Robustness evaluation of prognostic immune signature (PIS) in predicting progression-free survival. (**A**): 18 genes constituting PIS and their corresponding risk coefficient, *p* value, and hazard ratio. The sum of the expression level of each gene multiplied by its corresponding coefficient is the PIS score. The forest plot on the right visualizes the hazard ratio of the marker. If the hazard ratio of a gene is less than 1, it is indicated in blue, otherwise, it is indicated in red. In addition, the line length represents the 95% confidence interval of the hazard ratio. (**B**,**C**): The AUC of 1-, 2- and 3-year progression-free survival in the training cohort and validation cohort. The 1-, 2- and 3-year AUC for PFS was 0.940 (95% CI: 0.894−0.985), 0.981 (95% CI: 0.960−1.000), and 0.969 (95% CI: 0.937−1.000) in the training cohort, respectively. With 1-, 2- and 3-year AUCs of 0.816 (95% CI: 0.681−0.950), 0.865 (95% CI: 0.754−0.977), and 0.827 (95% CI: 0.699−0.955) in the validation set, respectively.

**Figure 8 ijms-24-05332-f008:**
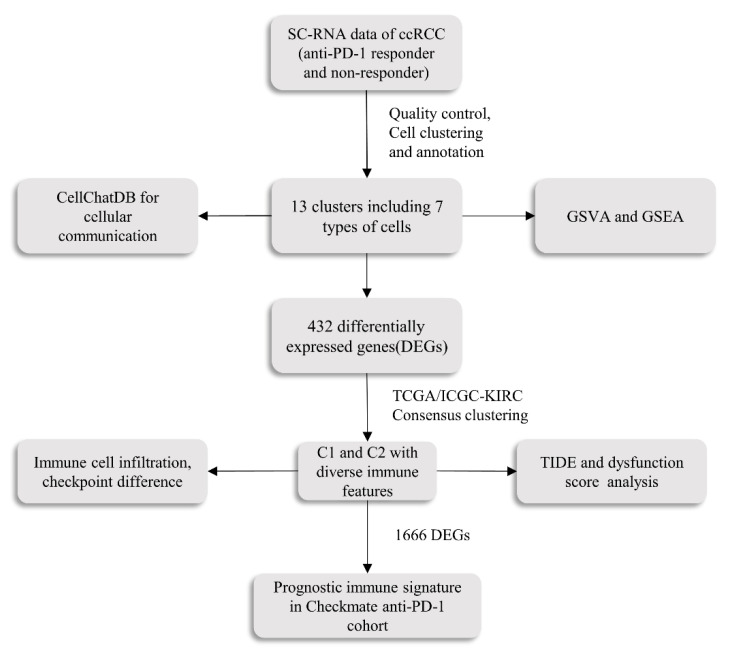
The overall workflow of the study. The scRNA-seq data were obtained from two patients who underwent cytoreductive nephrectomy after anti-PD-1 treatment (one was responsive to nivolumab and the other was nonresponsive), the prognostic immune signature (PIS) was established and verified in 172 patients (133 non-responders, 39 responders) with anti-PD-1 treatment. SC-RNA, single-cell RNA; ccRCC, clear cell renal cell carcinoma; GSVA, Gene Set Variation Analysis; GSEA, Gene Set Enrichment Analysis; TIDE score, Tumor Immune Dysfunction and Exclusion (TIDE) score.

## Data Availability

The datasets presented in this study can be found in online repositories. The names of the repository/repositories and accession number(s) can be found in the article/Appendix A.

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
