# Peer review of "A New Signature That Predicts Progression-Free Survival of Clear Cell Renal Cell Carcinoma with Anti-PD-1 Therapy"

_ijms, 2023, doi:10.3390/ijms24065332_

Round 1

Reviewer 1 Report

There are few new biomarkers to identify molecular subtypes of renal clear cell carcinoma and predict survival time after anti-PD-1 treatment. In this study, by combining single-cell RNA seq with Bulk-RNA seq, the authors constructed prognostic model and validated triumphantly to predict progression-free survival

in ccRCC receiving ICI treatment. They found that the expression level of PDCD1/PD1 is not an effective marker to predict the response to ICIs; moreover, they constructed prognostic immune signature (PIS), which divided ccRCC patients with anti-PD-1 therapy into low-risk and high-risk group. The PFS and immunotherapy response rate of low-risk group were significantly better than those of high-risk group. In summary, they established a robust PIS to predict the progression-free survival of ccRCC patients receiving immune checkpoint inhibitor. These results are very interesting, and the study is well designed.  

Author Response

Dear Professor:

Thanks very much for your kind work and consideration of our paper. On behalf of my co-authors, we would like to express our great appreciation to you and editor.

Reviewer 2 Report

In this research, the authors tackled the issue of anti-PD-1 immunotherapy being inefficient to improve survival time and quality of life of patients with clear cell carcinoma (ccRCC). Using single-cell RNA-seq datasets from previous studies, they have re-analyzed and extracted populations of CD4+T and CD8+T to observe cell signalizing pathways involved in the disease treatment. They found through bioinformatics analysis (GSVA, GSEA, and cell-cell interactions) that PDCD1/PD1 is not an effective marker to predict the response to Immune Checkpoint Inhibitors at the current models. Moreover, their approach to construct Prognostic Immune Signature (PIS) divided ccRCC patients into two groups with better clinical prognosis. I believe this research is of a great interest for the scientific community and medical area. However, I am afraid I have some major points that should be taken into consideration before this results become public.

Major issue

1.     Although authors claim that they have re-analyzed and processed scRNA-seq. I think their analysis is problematic and lead to wrong assumptions. For instance, in the UMAP in Figure 2 A, there are 7 clusters identified. However, a cluster annotated as C2-RM CD8+T is rather a reasonable representation of a unique cluster. We can observe that such a cluster is divided into three, with totally different transcriptome profiles. For example, the LAG3 gene expression shown in Figure 3A provides evidence that EX CD8+T, C2-RM CD8+T, and Cytotoxic CD8+T represent a similar gene expression of cells on that top clusters. But the proportion of C2-RM CD8+T connected to Treg CD4+T may have a totally different transcription profile (see gene CTLA4). That being said, I wonder why the authors kept the analysis of such weird clustering for further insights. I would suggest the author consider a range of resolutions in the FindClusters function of the Seurat pipeline (c(0.1, 0.2, 0.3, 0.5, 0.6, 0.8, 1)) and choose an appropriate value.

2.     The prediction values are very high, and the number of samples using to predict is low. How did the authors address the problem of overfitting?

 Minor issues

1.     In Figure 2, clusters are represented in multiple colors for the same cell type. Please remake the plots with the same color for each cluster. For example, Figure 2A EX CD8+T is grey but in Fig.B it is red.

2.      English language improvement is necessary.

Example:

“Procession” should be processing.

Several commas missing, and misplacements.

Capital letters are used wrongly throughout the manuscript.

Reviewer 3 Report

Authors report a signature to predict response to immunotherapy in patients with Clear Renal Cell Carcinoma. Although the AUC in the training and validation sets are good, there are some important questions that need to be addressed.

Major points

·       - A very important detail that has not been explained or taken into account is the nature of the scRNA-seq experiment. Some very important questions have not been explained:

1.       How many patients are included in the sequencing?

2.       How many cells were sequenced?

3.       How many tumor versus non-tumor cells were sequenced?

4.       Were the cells collected prior to immunotherapy? This is a very important point. If your aim is to derive a predictor that would be analyzed prior to therapy in order to decide who is going (or not) to receive immunotherapy, then you have to derive your predictor from treatment-naïve patient samples. Are the patient samples in the scRNA-seq experiment collected prior to immunotherapy? If not, it would be very difficult to justify the use of this data to predict response prior to therapy.

·       This information is very important to understand whether the signature obtention was performed following a proper rationale.

·       - PIS predictor: as the predictor was built from a cohort with PFS and OS outcomes after treatment, authors should show how the predictor works using OS as endpoint. Please provide the OVERALL SURVIVAL plots in the training and validation sets, as well as the AUC curves and values.

·     -   PIS predictor: please proved the sensitivity and specificity of the predictor, as these 2 values would be very important for clinical decisions.

·       - How is the PIS predictor in the same CheckMate cohort when compared with others, such as  those of references 74 and 75?

Minor points:

·       - Nivolumab, pembrolizumab, etc (see intro) are not PD1/PDL1 inhibitors. They are antibodies. Please correct.

·       - Figure legends should be more informative, so the reader can understand the graphs and panels without going back and forth to M&M and results section.

·       - Figure 6. Please write on survival plots the end-points PFS or OS. Please provide the survival plots of the cohorts on panel C as supplementary info

·       - References within the text should not be PMID…, neither GSE… If the results have been published in journals, please refer them with “Surname et al” or similar formulas.

·       - Please, explain more details of the different cohorts in the M&M section. I would include the information that is necessary in the context of the manuscript.

Author Response

Please download the attachment to see response to reviewers  3.

Round 2

Reviewer 2 Report

The authors have improved the manuscript during the revision process. Moreover, they could answer my questions appropriately and included some of my suggestions. I have a few concerns but the points were included in the manuscript so the readers will have that in mind. 

Author Response

On behalf of my co-authors, We would like to thank the referee again for taking the time to review our manuscript.

Reviewer 3 Report

The authors have answered correctly to the points raised.

Before publishing, english language, style and grammar need a very thorough revision. Only in the abstract there are 4-5 grammar mistakes.
